# Safety and Efficacy of Sildenafil for Group 2 Pulmonary Hypertension in Left Heart Failure

**DOI:** 10.3390/children10020270

**Published:** 2023-01-31

**Authors:** Kinjal Desai, Michael Di Lorenzo, Warren A. Zuckerman, Ezinne Emeruwa, Usha S. Krishnan

**Affiliations:** 1Division of Pediatric Cardiology, New York Presbyterian Morgan Stanley Children’s Hospital, New York, NY 10032, USA; 2Columbia University Vagelos College of Physicians and Surgeons, New York, NY 10032, USA

**Keywords:** pulmonary hypertension, sildenafil, precapillary, heart failure, echocardiography

## Abstract

Pulmonary hypertension (PH) is a multifactorial, progressive disease with poor outcomes. Group 2 PH is defined by pulmonary vascular disease with elevated pulmonary capillary wedge pressure including both left-sided obstructive lesions and diastolic heart failure (HF). Sildenafil was historically discouraged in this population as pulmonary vasodilation can lead to pulmonary edema. However, evidence suggests that sildenafil can help to treat the precapillary component of PH. This is a single center, retrospective pilot study of pediatric PH patients with left-sided HF who were treated with sildenafil for ≥ 4 weeks. HF patients without mechanical support (HF group) and HF patients with a left ventricular assist device (HF-VAD) were analyzed. The exploratory analysis described the safety and side effects of the drug. Echocardiographic parameters were compared before and after sildenafil treatment in a paired analysis. The changes in medical therapy during treatment, mechanical support, and mortality was reported; 19/22 patients tolerated sildenafil. Pulmonary edema in two patients resolved upon discontinuation of sildenafil. In the HF group, both the right atrial volume and right ventricular diastolic area decreased, and the tricuspid regurgitation (TR) S/D ratio decreased after therapy (*p* = 0.02). Across both the groups, four patients weaned off milrinone and seven weaned off inhaled nitric oxide. Of the thirteen HF patients, four received a transplant, and all of the nine HF-VAD patients received a transplant. Sildenafil can be safely used in carefully selected patients with HF and mixed pre/postcapillary PH with judicious titration and inpatient surveillance, with patients showing improvements in echocardiographic parameters.

## 1. Introduction

Pulmonary hypertension (PH) is a multifactorial and progressive disease that has poor outcomes without treatment. PH is defined by a mean pulmonary artery pressure (mPAP) of >25 mmHg in adults [1]. This definition was updated at the 6th World Symposium for Pulmonary Hypertension (WSPH) in 2018 in Nice, France to specify the definition of pulmonary arterial hypertension (PAH) as a mPAP > 20 mmHg and to include pulmonary vascular resistance (PVR) > 3 Wood Units and pulmonary capillary wedge pressure (PCWP) <15 mmHg [1,2,3]. On the other hand, postcapillary disease in pulmonary hypertension is defined by an elevated PCWP as the cause of PH [2]. The difference between these two components of PH often guides clinical management. However, it is well known that a number of patients with Group 2 PH also have a precapillary component to their disease which should be considered for optimal management.

Per the WSPH6, PH can be categorized into five distinct clinical groups based on the physiology or phenotype of the PH (Figure 1) [4,5]. Pediatric patients with congenital heart disease mostly fall into Groups 1, 2, and 5. Group 1 includes patients with simple congenital heart disease (CHD) with Eisenmenger physiology, PAH with left-to-right shunts, PAH incidental to CHD, and postoperative defects [2]. Group 5 includes patients with complex CHD, including a single ventricle and tetralogy of Fallot with pulmonary atresia and major aortopulmonary collaterals. 

Group 2 PH includes patients with pulmonary vascular disease, elevated PCWP, and a mix of post precapillary and postcapillary components. Cardiomyopathy, including left ventricular (LV) systolic/diastolic dysfunction and congenital left heart inflow/outflow obstructions and pulmonary vein obstruction, are the two major types of patients in Group 2 PH [2]. Adults with mixed PH have experienced considerable benefits from sildenafil therapy in targeting the precapillary component of disease and in independently helping with LV remodeling [6,7,8]. It has not been used frequently in pediatrics due to concerns around pulmonary edema. 

Our study aims to retrospectively assess the safety and efficacy of sildenafil for the treatment of PH in Group 2 PH patients with heart failure (HF).

## 2. Materials and Methods

This study was a single center, retrospective pilot cohort study. All the pediatric patients followed at Columbia University Irving Medical Center/Morgan Stanley Children’s Hospital of New York with pulmonary hypertension who were treated with sildenafil between the years of 2007 and 2020 were assessed for candidacy. The inclusion criteria were patients with Group 2 PH secondary to heart failure with or without a left ventricular assist device (LVAD). PH was diagnosed prior to the start of any PH therapy, including sildenafil in all the patients, by a combination of echocardiographic, hemodynamic, and clinical criteria. From the original population, patients with Group 1 and 5 PH were immediately excluded. The patients with left-sided obstructive lesions in Group 2 PH were excluded given the relative heterogeneity of the disease, including the type of lesions, the number of interventions, and the severity of the lesion. The patients with right ventricular assist devices (RVADs) were excluded in order to remove confounding variables. Patients post heart transplant and patients with post-operative PH with right ventricular (RV) failure were not included in the study. 

The study design and timeline is summarized in Figure 2. The patients were divided into two groups and were analyzed independently. The HF group consisted of patients with left heart failure (LHF) without mechanical support. The HF-VAD group consisted of LHF patients with LVAD support. The mechanically supported patients were analyzed separately from the non-mechanically supported patients because the baseline hemodynamics and the physiology of heterogenous left heart support was too discrepant between the groups. For accuracy and in order to minimize the confounding factors, the groups were kept separate. The pre-sildenafil echo was defined as the most recent echo prior to the start of sildenafil and after any major changes in physiology. For the patients with an LVAD or those who received any catheterizations or surgical interventions, the pre-sildenafil echo was isolated after these changes. Sildenafil therapy was required consistently without lapse or noncompliance for a minimum of 1 month and a maximum of 4 months. The post-sildenafil echo was obtained 1-4 months after sildenafil therapy and the patients were followed for their clinical outcomes until their latest follow-ups. 

Patients who started sildenafil treatment but underwent implantation of mechanical support or a transplant before a minimum of 1 month of therapy were excluded from the study. Any patient who started sildenafil treatment but stopped secondary to side effects or clinical decompensation were included and reported. 

The primary aim to assess the safety of sildenafil was reported in an exploratory analysis of the clinical side effects at the start of drug therapy. The primary outcomes used to assess the efficacy of sildenafil were the echo parameters of RV function and PH, as outlined by our institution’s PH echo protocol. The secondary outcomes included clinical descriptive parameters, including the degree of respiratory support, the amount and type of inotropes, clinical changes, mechanical support, and heart transplant/survival. 

The echocardiograms were isolated in two major study points: pre-sildenafil and post-sildenafil. These patients’ echocardiograms were isolated, and the institutional PH echocardiogram protocol was applied retrospectively to all the studies by a single standardized physician in all cases. All the measurements were performed using Syngo Dynamics software with a reviewer who was blinded to the demographic information of the subjects and previous patient echo measurements. A second reviewer, also a pediatric cardiologist, reviewed a subset of the studies and was blinded to the patients’ demographics, the study groups, and the phase of the study. When compared, there was sufficient inter-reader reliability of the echo measurements between both the reviewers. 

The echocardiographic measurements were obtained according to the standards of the American Society of Echocardiography (SITE) by both the readers referenced below [9]. If adequate images were not present in these retrospectives, the measurement was not taken and the paired analysis on that parameter was not conducted. 

The echocardiographic measures included the right atrium (RA) area, volume in the major and minor axes, right ventricle (RV) systolic and diastolic area, fractional area change, and two-fold length and height. Tricuspid regurgitation (TR) dopplers with velocity, an estimated gradient, and a systolic/diastolic ratio (TR S/D ratio) were recorded, as well as E and A waves and an E/A ratio. This echocardiographic data was obtained before sildenafil treatment only; however, additional PH therapies may have already been started, as discussed below. Therefore, it is important to note that prior to the start of any of the PH therapies in the HF group, given the degree of LV failure, the RV/LV systolic pressure ratio was estimated at >50% for all the patients. This could be not reported in the HF-VAD group with accuracy given the presence of mechanical support. 

The tissue doppler (Td) for the RV lateral wall was recorded with an RV Td myocardial performance index (MPI). When available, the RV M-mode tricuspid annular plane systolic excursion (TAPSE), LV eccentricity index (LVEI), and LV ejection fraction (EF) and/or shortening fraction (SF) were recorded. All the echocardiographic measures recorded are well-published parameters used to evaluate PH in the echocardiography literature [10,11,12,13,14,15]. 

Clinical data regarding respiratory support, medications, and descriptive outcomes were all gathered via retrospective chart review. The chart review was a mix of both electronic medical records and paper charts that were scanned into the medical record. Respiratory support was collected retrospectively from intensive care unit notes, as were the medications given at the start of sildenafil therapy, during therapy, and at the end of the study period. The respiratory support information included invasive versus noninvasive ventilation, the amount of fractional inspired oxygen, and the use of inhaled nitric oxide. For the medications, the number of diuretic agents, the number of vasoactive medications, and the type (milrinone, anti-arrhythmic medications, and cardiac medications for afterload reduction) were all recorded. Regarding the imaging data, chest X-rays were assessed the day after the start of sildenafil treatment and for the next two subsequent chest X-rays. The goal was to follow an objective measure of pulmonary edema on imaging. The reports used were only those from radiologists included in the formal study results. Cardiac catheterizations were also assessed for descriptive outcomes. Hemodynamic data recorded in formal catheterization reports were collected for those few patients who did have catheterization data either pre- or post-sildenafil without any other invasive or surgical interventions. Finally, the patients were followed through to the latest follow-up known in our hospital system in order to record the need for mechanical support, as well as overall survival or mortality rates. The causes of mortality were also recorded and reported. 

Baseline and demographic characteristics of both the study groups were independently summarized as continuous variables using means and SDs or medians and the interquartile range (IQR). For the categorical variables, the data was expressed as percentages. The echocardiographic outcome variables were summarized predominantly via the median and IQR given the asymmetric distribution of the data in both the groups. The paired analysis was performed for each patient and was compared to himself/herself before and after sildenafil use. A Wilcoxon or Mann–Whitney rank sum test was used to analyze the means of the continuous variables pre- and post-sildenafil use. A two-sample *t*-test was not reported given the non-normal distribution of the data sets. A *p*-value < 0.05 was considered statistically significant. 

Given the fundamental difference in physiology between non-mechanically supported patients and mechanically supported patients, these two patient groups were analyzed separately, with comparisons only for the demographic and baseline variables, including age and the time of sildenafil therapy.

## 3. Results

A total of 22 patients were isolated for the study, 13 of which were in the HF group and 9 were in the HF-LVAD group. The patient demographics are shown in Table 1 for both the groups. In the HF group, 8/13 (61.5%) were female vs. 4/9 (44.4%) in the HF-LVAD group. The etiology of acute HF ranged from primary cardiomyopathy (73%), LV diastolic dysfunction secondary to rejection of a transplanted heart (14%), chemotherapy-induced cardiomyopathy (4%), and LV dysfunction secondary to myocarditis (9%). Congenital structural heart disease was present in 5/22 of patients. Prior to the study period, all five patients were either fully repaired with no residual lesions or had been transplanted remotely. A total of 6/22 (27%) of patients had a known genetic disorder, 5 of which were in the HF group and 1 was in the HF-LVAD group. In the HV-LVAD group, 3/9 (33%) were supported with Heartmate II, 2/9 (22%) were supported with Heartware, and 4/9 (45%) were supported with Berlin. 

### 3.1. Drug Safety

Of the 22 patients, 19 remained on sildenafil for a therapeutic amount of time and 2 stopped sildenafil secondary to worsening pulmonary edema (1 patient from the HF group and 1 patient from the HF-LVAD group). Furthermore, one patient from the HF group stopped sildenafil secondary to an inability to take oral medications. An additional two patients, one in each study group, stopped sildenafil after experiencing worsening pulmonary edema; however, they were able to diurese and restart therapy within 1–2 days with good subsequent tolerance of the drug, thereby completing the study.

### 3.2. Echocardiographic Data

The echocardiographic results of the 2D measurements relating to right heart size, RV area change, and TAPSE are shown in Table 2. Overall, in the HF group, both the RA area and the RV end diastolic volume trended down with sildenafil use in the paired analysis, but did not show a statistically significant change. Additionally, the RV/LV end-systolic ratio and LVEI remained unchanged overall before and after sildenafil use. Similarly, in the HF-LVAD group, the 2D echo parameters showed no statistically significant changes with sildenafil use.

Table 3 details relevant doppler data including TR doppler measures as well as assessment of tissue doppler. The median TR velocity was 3.09 m/s and 2.06 m/s in the HF and HF-VAD groups respectively. In the HF group, TR S/D ratio showed statistically significant decrease in paired analysis after therapeutic use of sildenafil. Other doppler parameters across both the HF group and the HF-LVAD group showed overall no measurable pattern of change.

### 3.3. Additional Secondary Outcomes

Descriptive data regarding the additional secondary outcomes were also collected and reported when available. Of the 19 patients who completed sildenafil therapy, 11 patients in the HF group and no patients in the HF-VAD group had cardiac catheterizations pre-sildenafil. For the 11 patients in the HF group, the median for mean PA pressure was 32.5 mmHg, with a median pulmonary capillary wedge pressure of 21 mmHg and an indexed pulmonary vascular resistance (PVRi) of 4.35 WU m^2^. Post-sildenafil catheterization showed a mean PA pressure of 28 mmHg, a mean pulmonary capillary wedge pressure of 21 mmHg, and a mean PVRi of 5.3 WU m^2^. Overall, the pulmonary vascular resistance indexed (PVRi) reported ranged from 2 to 18 WU m^2^, with no consistent trend. Of note, no patient had all the hemodynamic parameters measured and reported. Therefore, the medians were reported for just those available. In total, in the HF group, only seven patients had both pre-sildenafil and post-sildenafil heart catheterization data performed on varying degrees of support and both at home and outside institutions. Given the heterogeneity of the studies and the paucity of numbers, the paired comparison was not conducted. 

Regarding additional medications, in the HF group alone, no patients were on continuous vasopressors prior to starting sildenafil, four out of eleven were on digoxin, five were on a beta blockade, four were on enalapril, and all eleven patients were on diuretics. One patient was also on bosentan. Significantly, five patients were on milrinone at the start of the study. Throughout the study period, the additional medications started were epinephrine (one patient), iloprost (one patient), digoxin (two patients), and milrinone (one patient). Post-sildenafil therapy, three patients were able to wean off of milrinone. In the HF-LVAD group, at start of sildenafil therapy, five patients were on at least one vasoactive medication, such as epinephrine or dobutamine, digoxin (three patients), amiodarone (three patients), nitroprusside (three patients), and no patients were on enalapril, whereas all the patients were on milrinone and diuretics. During the study period, one patient started digoxin and one started enalapril. 

Collectively, across both the HF and HF-LVAD groups, four patients weaned off of milrinone while on sildenafil therapy. Additionally, all seven patients who were on inhaled nitric oxide at the start of the study were weaned off. With the exception of only 1 patient, all the other 18 patients either remained on the same level of respiratory support or were weaned with sildenafil therapy, whereas all the elevated FiO_2_ requirements were weaned to 21%. The first three chest X-rays were recorded starting the day after beginning sildenafil therapy for every patient. Of the 22 patients, 2 did not have surveillance films while they were inpatients, and 2 were started on sildenafil treatment as outpatients. Of the remaining 18 patients, the 2 who had to temporarily halt sildenafil for a few days before restarting both showed slightly worse pulmonary edema on chest X-ray that resolved quickly with diuretics. The two patients who stopped sildenafil, given their worsening clinical conditions, also showed significant pulmonary edema, effusions, and atelectasis 3 days after sildenafil, which resolved after stopping the medication. Of the remaining patients who continued sildenafil therapy, almost all showed mild edema or small effusions on chest X-ray 1 day after therapy, which either improved or fully resolved within 3 days. The outcomes in both the HF and HF-LVAD groups, including the need for mechanical support, transplantation, death, and last follow-up, are expressed in Figure 3. The reasons for stopping sildenafil for patients in both the groups are shown in Figure 3.

## 4. Discussion

Group 2 PH in some patients with LHF can present with an additional element of pulmonary vascular disease and an elevated PCWP, a mix of post precapillary and postcapillary components. Left-sided filling pressures are elevated and subsequently transmitted to the pulmonary venous and arterial vasculature [16]. Group 2 PH in LHF may initially start as ‘isolated postcapillary disease’ occurring distal to the small pulmonary arterioles without intrinsic vascular abnormalities. However, as PH is a dynamic system, chronically elevated PAP eventually leads to arteriolar muscular hypertrophy and some vascular reactivity, constituting precapillary disease [17]. This mix of pre- and postcapillary vascular disease is often what makes the treatment of PH in LHF more challenging. 

Sildenafil is a phosphodiesterase 5 inhibitor that has been used to treat childhood PAH since the early 2000s when it was approved in adults [18]. Conventional understanding suggests that, in Group 2 PH, pulmonary vasodilation in the setting of an elevated PCWP will flood the precapillary vasculature and lead to pulmonary edema. However, recent clinical studies in adults have shown some benefit in using sildenafil for PH in left heart disease by targeting the precapillary component of PH and by possibly causing LV remodeling [19,20,21,22]. Studying the effects of sildenafil on Group 2 PH patients may allow us to consider a new use for sildenafil in pediatric LHF patients. 

In this single center pilot retrospective study of pediatric LHF and HF-LVAD patients on sildenafil, we believe, overall, that sildenafil is safe to use. Of the patients in our study, 86% safely tolerated sildenafil therapy. Although therapy was briefly paused for two patients secondary to worsening pulmonary edema and desaturations, both patients returned back to baseline shortly after using diuretics and did not have a significant adverse outcome or morbidity related to sildenafil therapy. This suggests that sildenafil can be safely used in-house with careful titration, surveillance, and possibly escalated diuresis in order to prevent and treat early pulmonary edema. Additionally, given this study required a minimum continuous therapy of 1 month without escalation to mechanical support to alter hemodynamics, patients that were inherently excluded from the study include those who started sildenafil and tolerated it without side effects, but either received an LVAD or an orthotopic heart transplant (OHT) prior to the 4-week period or clinically improved enough to stop therapy. 

As mentioned previously, multiple studies have demonstrated that sildenafil therapy can improve PH in Group 2 adult patients with HF. One of our primary outcomes was to assess the efficacy of sildenafil therapy in this study by focusing on echocardiographic data, as there was a paucity of complete hemodynamic data before and after starting sildenafil. The echo parameters assessed here have been used at our institution to estimate the severity of PH or as a trackable marker of change in PH along with right/left heart remodeling. The 2D measures of RA volume and RV diastolic area both trended down in the HF group after sildenafil therapy. This trend may suggest that, should more patients be analyzed or if therapy perhaps was for a longer duration, both the RA volume and RV end diastolic area would decrease as the right heart remodels. 

Our study showed that the TR S/D ratio significantly decreased after sildenafil therapy (*p* = 0.02), specifically in the HF group. The TR S/D ratio has been shown in multiple studies to be a strong marker of survival for children with PH. Current studies report a TR S/D ratio above 1.4 to inversely correlate with survival [23]. In this study, the TR S/D ratio significantly reduced from a mean of 1.37 to 1.14. Studies have shown that every increase of 0.1 in the ratio correlates with a 13% increase in the yearly risk of lung transplant or death for pediatric patients [23]. Although this parameter cannot judge the efficacy of sildenafil alone, larger and long-term studies may, in the future, be able to show the benefits of PH treatment and may assess changes in survival/mortality, if any. 

Using advanced echocardiographic Doppler parameters, no worsening was seen between pre- and post-sildenafil studies in either of the study groups. Many of the patients in our study were hospitalized during the therapy, which speaks to the selection bias in this group. As we are more cautious to begin this therapy in general and are more likely to start it in already hospitalized patients, we may be inadvertently selecting a more severe group of HF patients in whom showing an effective change in disease may be difficult. Another very important challenge to showing positive echocardiographic changes in this population is the progressive nature of the disease in pediatric cardiomyopathy and HF.

The descriptive data points we collected, including a change in medications, respiratory support, and radiologic evidence of pulmonary edema, were very helpful in understanding the role of sildenafil clinically. Given that, overall, most patients on milrinone and/or inhaled nitric oxide were able to wean off these medications after starting sildenafil suggests that sildenafil may be a safe adjunctive oral medication to help patients transition off of other pulmonary vasodilators and may help to de-escalate care. 

The limitations of this study include the single center patient base. This patient population is very specific, extremely sick, and, as expected, a small group. The retrospective format of this data reflects the institutional trends in medication decision making. In this sense, a wider multi-institutional study may shed better light on the true efficacy of this drug. Additionally, as this was a retrospective study, much of the data for the paired analysis for the pre- and post-therapy phases were limited due to a lack of adequate images in shorter point-of-care echo studies. Finally, this study was likely underpowered. However, this was an anticipated challenge for this study given the relative uncommon use of sildenafil specifically in this pediatric population. Although the exclusion of CHD (consisting of left-sided obstructive lesions) did help with homogeneity within the population, the relative difference in age across each study group, along with the type of HF and the degree of required medical support when sildenafil was started, is still fairly varied and may not allow for the isolation of specific groups of patients who may benefit more from this therapy than others. In the future, this pilot study can be used as a model for investigating sildenafil in Group 2 PH in a larger prospective multicenter study.

Investigators studying mouse models with induced LHF have used sildenafil for PH treatment in mock Group 2 lesions and have shown attenuation of LV remodeling over time, which was a previously unknown benefit. Studies in rats with aortic valve insufficiency and mice with thoracic aortic banding showed significant T-tubule remodeling and improved LV contractile function with sildenafil [24,25]. These findings, together with adult literature in HF patients, seem to suggest the potential role for sildenafil in carefully selected Group 2 PH patients, both in terms of cardiopulmonary hemodynamics, but also with the previously unknown benefits of improving the markers of LV function and remodeling. Although not studied here, we hope that future studies may investigate the benefits of sildenafil in HF patients as it relates to function and LV remodeling.

## 5. Conclusions

Overall, sildenafil is safe to use and is well tolerated in selected pediatric patients with heart failure and mixed Group 2 pulmonary hypertension, in conjunction with escalated diuretic therapy. Close surveillance and expedited intervention to address pulmonary edema is necessary in the patients who are started on pulmonary vasodilators in Group 2 PH with LHF. Additionally, in our study, sildenafil decreased the TR S/D ratio in HF patients without mechanical support, suggesting a potential benefit of the treatment.

## Figures and Tables

**Figure 1 children-10-00270-f001:**
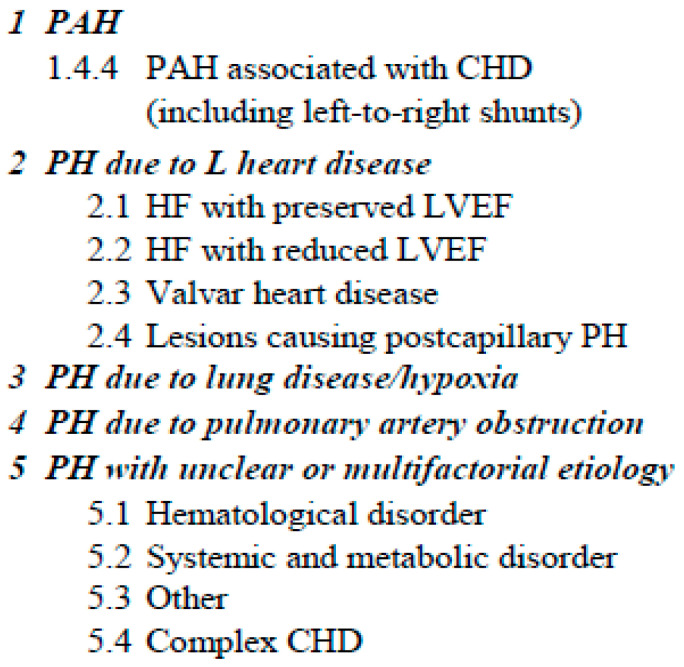
Classification of pulmonary hypertension per the 6th World Symposium on Pulmonary Hypertension (Adapted figure).

**Figure 2 children-10-00270-f002:**
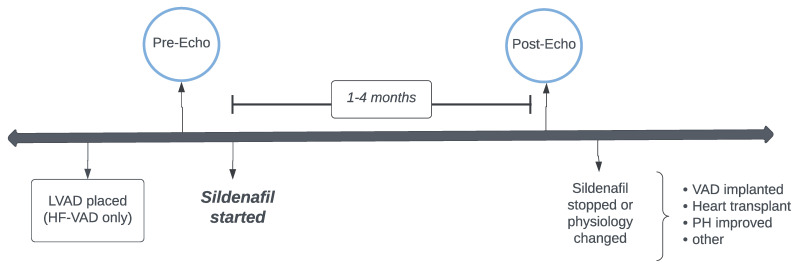
Patient study timeline.

**Figure 3 children-10-00270-f003:**
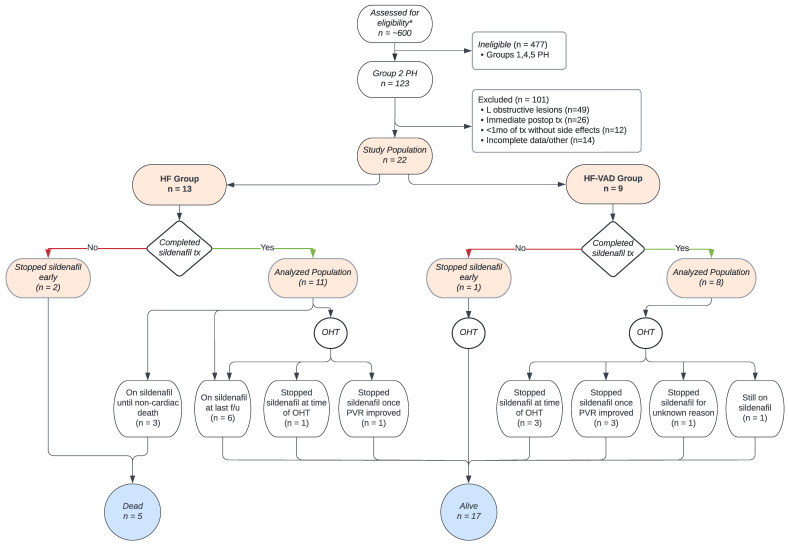
Patient selection, group allocation, and outcomes in the HF and HF-VAD groups. * Inclusion criteria: pediatric patients treated between 2007–2020 with Group 2 PH secondary to HF. Exclusion criteria: single ventricle/complex congenital patients, left-to-right shunts, lung disease, idiopathic PH, and Group 2 PH with left-sided obstructive lesions. Patients with right-sided mechanical support were excluded.

**Table 1 children-10-00270-t001:** Demographics. Congenital heart disease (CHD), restrictive cardiomyopathy (RCM), dilated cardiomyopathy (DCM), noncompaction cardiomyopathy (NCCM), hypertrophic cardiomyopathy (HCM), and orthotopic heart transplant (OHT).

	HF (*n* = 13)	HF - VAD (*n*= 9)
**Age**, months	111.4 (20.5–136.4)	149.8 (48.7–183.4)
**Female**, *n* (%)	8 (61)	4 (44)
**Prematurity**, *n* (%)	2 (15)	0
**Genetic syndrome**, *n* (%)	5 (38)	1 (11)
**Prior/repaired CHD**, *n* (%)	3 (23)	2 (22)
**Cardiomyopathy (CM)**, *n* (%)		
–RCM	3 (23)	0
–DCM	3 (23)	7 (78)
–NCCM	2 (15)	1 (11)
–HCM	2 (15)	0
–Nonspecific	3 (23)	1 (11)
**Reason for HF**, *n* (%)		
–Progressive primary CM	9 (69)	7 (78)
–Viral myocarditis	0	2 (15)
–Rejection s/p OHT	3 (23)	0
–Chemo-induced CM	1 (8)	0

**Table 2 children-10-00270-t002:** 2D echo parameters in the HF and HF-VAD groups before and after sildenafil therapy. The values are expressed as medians (interquartile ranges) unless otherwise specified.

	HF (*n* = 11)			HF-VAD (*n* = 8)		
	Pre-Sildenafil	Post-Sildenafil	*p* Value	Pre-Sildenafil	Post-Sildenafil	*p* Value
**RA volume, ml**	28 (22–45)	25 (14–38)	0.28	36 (23–52)	28 (13–73)	0.67
**RV end diastolic area, cm^2^**	16 (8–19)	15 (9–16)	0.31	23 (14–30)	20 (13–23)	0.34
**RV fractional area change, %**	26% (19–31%)	23% (20–32%)	0.92	24% (17–30%)	25% (22–30%)	0.6
**Tricuspid annular plane systolic excursion (TAPSE), cm**	1.6 (1.5–1.7)	1.8 (1.1–2.3)	0.63	0.74 (0.54–0.95)	1.16 (0.62–1.28)	0.19
**End-systolic RV/LV ratio**	0.68 (0.55–0.78)	0.68 (0.56–0.83)	0.86	0.70 (0.55–0.91)	0.49 (0.44–0.97)	

**Table 3 children-10-00270-t003:** Doppler echo parameters in the HF and HF-VAD groups before and after sildenafil therapy. The values are expressed as medians (interquartile ranges) unless otherwise specified.

	HF (*n* = 11)			HF-VAD (*n* = 8)		
	Pre-Sildenafil	Post-Sildenafil	*p* Value	Pre-Sildenafil	Post-Sildenafil	*p* Value
**TR velocity, m/s**	3.09 (2.38–3.44)	2.62 (2.36–3.93)	0.94	2.06 (1.95–2.32)	2.37 (2.21–2.61)	0.06
**TR S/D ratio**	1.37 (1.25–1.55)	1.13 (1.06–1.31)	** *0.02* **	1.42 (1.2–1.85)	1.37 (1.28–1.49)	0.82
**Td RV lat E′/A′ ratio**	1 (0.72–2.45)	0.71 (0.6–1.28)	0.29	0.9 (0.66–2)	0.77 (0.52–1.19)	0.73
**Td RV MPI**	0.58 (0.52–0.72)	0.59 (0.46–0.60)	0.49	0.79 (0.58–0.95)	0.51 (0.47–0.56)	0.11
**MV E/A ratio**	2.34 (1.8–3.09)	1.9 (1.5–3.3)	0.29	1 (0.78–1.64)	1.29 (0.82–1.7)	0.48

## Data Availability

Data supporting the reported results is unavailable due to privacy and ethical restrictions.

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
