# Peer review of "Safety and Efficacy of Sildenafil for Group 2 Pulmonary Hypertension in Left Heart Failure"

_children, 2023, doi:10.3390/children10020270_

Round 1

Reviewer 1 Report

Overall the study is well written and the results demonstrate the safety and efficacy of sildenafil for the treatment of pulmonary hypertension (PH) in Group 2 PH pediatric patients with heart failure.

All abbreviations should be defined at their first appearance in the,  text for example HF line 59.

In the results section, the authors present information about x-ray but this exploration is not described in the material and methods section.

Reviewer 2 Report

Comments for Authors

As authors mention in this paper, there are still many problems with the treatment for Group 2 PH. Though I think it is an important message that authors could use sildenafil safely for Group 2 PH patients, there are some problems in this paper that should be reconsidered.

1.      It is necessary to clarify the kind of examination method and criteria used to diagnose pulmonary hypertension if authors include the patients with Group 2 PH in the analysis.  Judging from the median TR velocity of 3.09 m/s at Pre-Sildenafil period in HF group and that of 2.06 m/s in HF-VAD group in Table 3, it seems that most of patients didn’t have complicated PH.  To examine the effect of sildenafil for PH of Group2 patients, those patients without PH should be excluded. 
If authors try to show that sildenafil is effective for the left-side heart failure patients with or without PH, both inclusion criteria and primary outcome are unsuitable for this study.

2.      I think it is important to investigate whether changes in pulmonary artery pressure are due to left atrial pressure or pulmonary vascular resistance in order to evaluate the effect of PH drugs in patients with Group 2 PH. Therefore, it is better to investigate the data of patients who underwent catheterization before and after the administration of sildenafil even if the number is small. For similar reasons, systolic and diastolic ventricular function of left ventricle is also important and should be evaluated before and after the medication.

3.      It is difficult to conclude from the current analysis that decrease of TR S/D ratio due to sildenafil usage suggests potential survival benefit because other parameters representing right ventricular function did not improve. In addition, the function of right ventricle including TR/SD ratio can also improve with increased diuretics, so I don’t think the results about the function of right ventricle shown by the authors were necessarily due to sildenafil.

4.      I think it is difficult to analyze the effect of sildenafil for the function of right ventricle or the clinical courses of patients with Group 2 PH because there are so many factors affecting the condition of these patients. It seems necessary to compare the groups of patients taking and not taking sildenafil under similar conditions.

Round 2

Reviewer 2 Report

The corrections in response to reviewers were appropriate. Each comment for reviewer's questions was also appropriate.